# Inertial Sensor Location for Ground Reaction Force and Gait Event Detection Using Reservoir Computing in Gait

**DOI:** 10.3390/ijerph20043120

**Published:** 2023-02-10

**Authors:** Sara Havashinezhadian, Laurent Chiasson-Poirier, Julien Sylvestre, Katia Turcot

**Affiliations:** 1Interdisciplinary Center for Research in Rehabilitation and Social Integration (CIRRIS), Department of Kinesiology, Faculty of Medicine, Université Laval, Quebec, QC G1V 0A6, Canada; 2Department of Mechanical Engineering, Interdisciplinary Institute for Technological Innovation, Université de Sherbrooke, Sherbrooke, QC J1K 2R1, Canada

**Keywords:** ground reaction force, gait event detection, inertial measurement unit, reservoir computing, sensor location

## Abstract

Inertial measurement units (IMUs) have shown promising outcomes for estimating gait event detection (GED) and ground reaction force (GRF). This study aims to determine the best sensor location for GED and GRF prediction in gait using data from IMUs for healthy and medial knee osteoarthritis (MKOA) individuals. In this study, 27 healthy and 18 MKOA individuals participated. Participants walked at different speeds on an instrumented treadmill. Five synchronized IMUs (Physilog^®^, 200 Hz) were placed on the lower limb (top of the shoe, heel, above medial malleolus, middle and front of tibia, and on medial of shank close to knee joint). To predict GRF and GED, an artificial neural network known as reservoir computing was trained using combinations of acceleration signals retrieved from each IMU. For GRF prediction, the best sensor location was top of the shoe for 72.2% and 41.7% of individuals in the healthy and MKOA populations, respectively, based on the minimum value of the mean absolute error (MAE). For GED, the minimum MAE value for both groups was for middle and front of tibia, then top of the shoe. This study demonstrates that top of the shoe is the best sensor location for GED and GRF prediction.

## 1. Introduction

Gait event detection (GED) and ground reaction force (GRF) predictions are essential for accurately monitoring gait patterns in individuals with and without musculoskeletal diseases (e.g., MKOA) [1,2]. These biomechanical metrics not only allow for the comparing the body’s mechanical deviations in MKOA with healthy gait [3,4], but may also identify the disease severity [5,6,7]. Traditional methods for GED and GRF measurements use bulky and expensive in-laboratory settings, including force plates embedded in the ground or in instrumented treadmills [8]. However, these methods are not only cumbersome and costly, but they also require dedicated space and a controlled environment (e.g., a motion analysis laboratory). Such a controlled environment may lead individuals to modify the walking strategies they would use in an ecological setting (e.g., gait speed variation [9]) [10,11].

To address this problem, several wearable sensors have been developed as an attractive alternative that could be used for direct/indirect measurement of these biomechanical metrics or even in the field for providing remote intervention (e.g., gait retraining) [1,2]. However, wearable sensors providing direct GRF measurements (e.g., load cells [12,13], pressure sensor arrays [14], and pressure insoles [15,16]) proved limited. Load cells are highly expensive and not practical. Pressure insoles and sensor arrays need calibration before each test and are unreliable. These wearable sensors are worn under the foot and thus modify the foot–ground interaction. They may even change the shoes rigidity that will modify the walking condition. Thus, inertial measurement unit sensors (IMUs), enabling the indirect measurement of these biomechanical metrics is a practical and cost-effective solution that is studied more and more [17,18].

For estimating GRF and GED using IMU data only, previous studies used methods based on biomechanical modelling (statistical or inverse dynamic approaches) or machine learning. Recently, Karatsidis et al. [19] used a set of 17 IMUs to estimate three-dimensional GRF and moments in gait using an inverse dynamic approach for eleven healthy individuals. Based on IMUs-obtained 3D acceleration signals (Acc), they established the kinematics of 23 segments; then, using Newton’s equation, the kinematics and the inertial properties of each segment, they determined the multidimensional GRFs. This approach was successful (the maximum root means square error [RMSE] for the frontal plane was 29.6%), but still required complicated musculoskeletal modelling and too many sensors (and their calibration) before use. Neugebauer et al. [20] suggested a different method based on a statistical modelling approach. They placed a biaxial accelerometer on the right hip to estimate GRF vertical component peak (pVGRF) in walking and running for thirty-five healthy teenagers. This study resulted in two statistical models: mixed and generalized regression, based on the significant relationship between the log-transformed pVGRF and features such as monitored acceleration, centre of mass, sex, type of locomotion (run), etc. The average absolute difference between the actual (measured with force plate) and predicted pVGRF for mixed and generalized models was 5.2% (±1.6%) and 9% (±4.2%), respectively. However, this study was limited because it could not predict GED and the temporal profiles of GRF due to the sensor type and its location. In addition, this approach was complicated further because it required the manual extraction of the input signal for statistical modelling.

The machine learning algorithms overcame the complexity of biomechanical modelling and data acquisition processes. For predicting GRF and GED in gait analysis with IMUs, artificial neural networks (ANNs) have already proved their usefulness because of their non-linear modelling and the input signal automatic extraction feature [21,22]. This method stemmed from the hypothesis that a relationship exists between the acceleration and the GRF/GED. Hence, Leporace et al. [23] were among the first to use an ANN with an IMU on the shank. The seventeen healthy individuals that took part in this study were instructed to walk on an instrumented treadmill at a self-selected walking speed. IMU 3D acceleration was used to estimate the multidimensional GRF. The error analysis comparing the estimated with the measured GRF showed an adequate prediction. However, the model did not investigate GED.

Ngoh et al. [24] suggested another ANN-based approach for GRF vertical component (VGRF) estimation via IMU anterior-posterior Acc on top of the shoe for seven healthy males running on a treadmill. This study also determined the stance phase via a previously established method [25]. The acceleration was then segmented into 100% data points of the stance phase. The ANN used the segmented Acc to estimate the VGRF; the results were then compared to the VGRF measured by force plates embedded into the treadmill. The average root mean square error was less than 0.017 of the participant’s body weight (BW). However, these results were limited to a small training sample that did not provide enough information to handle the input data variability. Moreover, this method first required the identification of the stance phase of each gait cycle before making a VGRF prediction. Resampling the data implied having at least one step delay before making a prediction. Therefore, the suggested sensor location and the network used in the Ngoh et al. study [24], did not necessarily yield good predictions that would work with a higher validation sample size and allow model fitting when variability arose.

Through an ANN approach, Gue et al. [26] investigated the best sensor location in walking in order to predict the GRF. The nine healthy young individuals that participated in this study were asked to wear instrumented shoes with 3D pressure insoles. The 3D Acc of the IMUs placed on C7, L5 and on the forehead helped to find the best sensor location. The authors found a relatively large difference in performance depending on the sensor location when in an uncontrolled outdoor environment. The results showed that L5 was the best location predicting VGRF with an average prediction error of less than 5.0%. The result could be attributed to the L5 location that is more stable and closer to the body’s centre of mass (COM). Indeed, a relationship exists between the translational acceleration and the GRF following Newton’s Second Law [27,28]. However, in case of aging and pathologies such knee osteoarthritis (OA), the upper body stability changes, and thus the results of sensor location in this study cannot necessarily be applied to other populations [29,30]. Furthermore, this study was limited to only VGRF sensor locations in healthy young individuals. For GED sensor location, a systematic review by Pacini Panebianco et al., 2018 [31], showed that sensors placed on the shank and the foot performed better regarding accuracy and reliability compared to lower trunk-based ones. However, no study yet exists for determining the best sensor location in an integrated task of GED and GRF prediction in healthy and MKOA individuals, which is required before an ecological intervention such as gait-retraining. Beside the gap in research for sensor location in the integrated task of GED and GRF, previous studies used different combinations of Acc for predicting these biomechanical metrics (uniaxial, biaxial, triaxial) [20,32,33], while the best Acc combination signal input is yet to be investigated.

To fill these gaps, an efficient and promising ANN must be used that can be quickly retrained for different locations and for a new type of temporal data input. Hence, Jaeger [34], suggested the reservoir computing (RC) recurrent neural network (NN) method. RC, as a recurrent NN paradigm, proved to be computationally low-cost and easy to be implemented in an IMU for further field out-lab interventions [35]. RC is specialized in processing temporal signals including accelerations, and is adaptable due to its fast and systematic training methodology that adjusts coefficients using a linear regression. The hardware implementation of RC can significantly reduce the scale factor and the energy needed to run such algorithms. In particular, a smart reservoir computing implemented in Micro Electro-Mechanical Systems (RC-MEMS) enables non-linear detection with an acceleration input [36].

The numerical implementation of RC called the echo states network (ESN), is one of the most important and used RC methods [34,37,38]. ESN [37], and has proved to be promising in biomechanics applications including gesture recognition [39], muscle drive-in actuation [40], exoskeleton control [41] and GED [42]. ESN previously demonstrated that sensors on the lower limb are excellent for GED prediction (MAE not more than 10 ms) [42]. However, the best sensor location for the prediction of integrated GED and GRF tasks using the same light reservoir computing algorithm as presented in [42], remains unaddressed. The novelty of this study is that it demonstrates the best sensor location for an integrated task of GED and multi-dimensional GRF prediction in able-bodied and MKOA individuals for the first time and via an ESN approach. For this aim, multiple training and testing datasets from different sensor locations on foot and shank were used to determine the best sensor location and nature of input signal associated with it.

## 2. Material and Methods

### 2.1. Participants

Twenty-seven able-bodied adults and eighteen MKOA individuals volunteered to participate in this study at the Center for Interdisciplinary Research in Rehabilitation and Social Integration (Cirris, Québec, QC, Canada). The participants’ demographic information appears in Table 1.

### 2.2. Inclusion–Exclusion Criteria

The healthy individuals were eligible if they were over 30 years old, had no knee pain or injury, no history of orthopedic surgery or diabetes, and no additional neurological disorders that impaired function. The patients were included if they presented medial knee OA confirmed with both clinical [43] and radiographic evidence on the Kellgren & Lawrence (KL) scale [44]. The MKOA individuals were excluded if they used a walking aid for short distances (cane, walker), had a recent history of lower limb surgery affecting their walking, a ligament rupture or cardiovascular and musculoskeletal disorders that limited their walking (e.g., back pain).

The ethical committee approved the study (#2021–2269) and all participants signed the informed consent form before participating.

### 2.3. Tasks and Procedures

To assess lower limb dominance, the healthy participants kicked a ball [45]. For the unilateral MKOA individuals, the targeted limb was the one with the affected knee and for bilateral MKOA individuals, it was the most painful side. Five IMUs (Physilog^®^4 and 5, Lausanne, Switzerland, 200 Hz) were placed on: (1) the top of the shoe (TS); (2) the heel (H); (3) above the medial malleolus (MM); (4) the middle and front of the tibia (MFT); and (5) the medial of the shank close to the knee joint (MK) (Figure 1). Participants self-selected their walking speed (SS) during a five-minute trial on the instrumented treadmill (Bertec, Columbus, OH, USA, 1000 Hz). Based on a previously established cross-correlation approach, the measured signals (IMUs and GRF) were synchronized in pre-processing [46]. The walking at three speeds was recorded: SS, 20% faster and slower than SS.

### 2.4. Echo State Network (ESN) Formulation

#### 2.4.1. Experimental Data Used in the Network

The inputs were considered as IMU Acc in AP, ML and V directions, and aimed at predicting the GRF and GED. The GRF and IMU signals were recorded independently. As for the predicted signal, the GRF signal used was in a vertical direction on the ground (V), tangential to the ground and perpendicular to the speed of the movement of the walker (ML), then tangential to the ground and parallel to the direction of the walker’s speed (AP), as shown in Figure 2. Figure 3 shows the GRF curves for healthy and MKOA individuals. The zones divided further to calculate the prediction error were the loading response phase, crucial for clinical analysis [47], between 0 and 18% of the gait cycle, and the pre-swing phase, between about 50 and 60% of the walk cycle.

The RC was trained individually on each of the five locations with one of the five combinations of acceleration axes. Table 2 presents the definition of the input vector **u**(t) for each combination (the first element of each vector, 1, is the bias value). With these five locations (Figure 1) and seven axes (Table 2), the task in predicting GRF is to evaluate the 35 different vector configurations of the input data.

#### 2.4.2. The Goal Data

The goal data introduced as $y_{p}^{t}\left(t_{i}\right)$ was a column vector of three GRF elements ($R\left(t_{i}\right)$). The signal was then normalized by the participant’s weight $W_{p}\left(N\right)$ (Equation (1)).
(1)$$\mathrm{for}\;\mathrm{LE},{\;y}_{p}^{t}\left(t_{i}\right)=\frac{1}{W_{p}}\left[\begin{array}{c} R_{\mathrm{AP}}\left(t_{i}\right)\\ R_{\mathrm{ML}}\left(t_{i}\right)\\ R_{V}\left(t_{i}\right) \end{array}\right]$$

For the GED, the $y_{p}^{t}\left(t_{i}\right)$ was a column vector of five binary elements.
(2)$$\mathrm{for}\;\mathrm{GED},{\;y}_{p}^{t}\left(t_{i}\right)=\left[\begin{array}{c} HS_{\mathrm{event}}\left(t_{i}\right)\\ {\mathrm{HP}}_{\mathrm{event}}\left(t_{i}\right)\\ \begin{array}{c} {\mathrm{FF}}_{\mathrm{event}}\left(t_{i}\right)\\ {\mathrm{TP}}_{\mathrm{event}}\left(t_{i}\right)\\ {\mathrm{TO}}_{\mathrm{event}}\left(t_{i}\right) \end{array} \end{array}\right]$$
where each element represents a binary indicator for the presence or absence of the following gait events: the heel strike (HS), the heel push (HP), the foot flat (FF), the toe push, (TP) and the toe off (TO) on the foot. For example, the index of HS_event was one for the time step representing the heel strike. Only one non-zero element was defined for each dimension of $y_{p}^{t}\left(t_{i}\right)$ per gait cycle. The gait events were defined using the measured experimentally and post-processed VGRF to identify the five gait events (Figure 4) in the stance phase as follows:HS: the rising edge of the VGRF.HP: the first maximal peak obtained after the HS.FF: the minimal peak obtained after the HP. For some walking patterns, especially for slow, as well as for MKOA participants, no minimal peaks in the middle of the stance phase were found. The FF was then defined by the middle point between HP and TP.TP: the last maximal peaks of the VGRF.TO: the falling edge of the VGRF.

The GED was performed using the peaks detected in the reservoir prediction output by a peak finder algorithm. The algorithm threshold of 1.06 s for healthy and 1.09 s for MKOA individuals was based on 0.7 times the patients’ mean gait cycle duration in the training sets. Each non-zero target index was compared to the closest peaks identified in the ESN prediction. The mean absolute error (MAE) for all events was independently computed by event and record.

#### 2.4.3. ESN Formulation for Finding the Goal Data

The ESN is a numerical implementation of the RC concept formalized by Jaeger [34]. Similar to other ANNs, ESN allows for the performing of a non-linear transformation between time-series Acc inputs u(t) to the goal prediction signal y(t) with an architecture schematized in Figure 5.

The input weight matrix W_in_ (N × N_in_) connects the input signal u(t) (N_in_ × 1) to the N neurons of a reservoir (which allows the linear combination of states to form a prediction signal y(t) (N_out_ × 1) (where Nout is the number of prediction signals). The function $f_{i}$ represents the activation function (tanh) of the reservoir ind node. In the case of a recursive tank, a feedback weight matrix W (N × N) connects the reservoir nodes with its past output states. The parameter is a weighting factor between the present state and the past state, which allows for the controlling of the forgetting rate of the past states u(t) to adjust the reservoir response dynamics. The iterative calculation of the ESN is:(3)$$\tilde{x}\left(t\right)=\tan h\left(W_{\mathrm{in}}u\left(t\right)+\mathrm{Wx}\left(t-\tau \right)\right)$$
(4)$$x\left(t\right)=\alpha \tilde{x}\left(t\right)+\left(\alpha -1\right)x\left(t-\tau \right)$$
(5)$$y\left(t\right)=W_{\mathrm{out}}x\left(t\right)$$
where y(t) represents a delay of 1 index in the past x states.

Matrix W_in_, W and the parameter α, were selected according to the ESN hyperparameters (HP) model according to the first optimization study published in [42]. HP were selected to optimize computing capability according to the CHARC metrics. The regularization parameter has been chosen similar to the one presented in [42] to minimize giving the best NRMSE in a validation dataset. The hyperparameters for reservoir computing are presented in Table 3, and they were established via the approach represented in [37], explaining how to build matrices W_in_ and W based on these hyperparameters.

### 2.5. Different Training Methods

This section presents the different training methods used in ESN: the standard and kernel methods.

#### 2.5.1. Standard Training Method

A unique Wout output weight matrix was established correlating with the exit value from the tank to the GRF and GED for all patients. We uniformly applied the RR method to all data (healthy and MKOA individuals). The database contained 135 series of continuous data (N_{TS} = 135) as the walking data collection of all participants. The training used the first 70% of each record trial duration. Therefore, the input data was defined based on Equation (6).
(6)$$u:\{U_{p}=\left[u_{p}\left(t_{1}\right),u_{p}\left(t_{2}\right),\ldots ,u_{p}\left(t_{i}\right),\ldots ,u_{p}\left(t_{T_{p}}\right)\right]|\forall 1<p<N_{\mathrm{TS}}\}$$

The number of training time steps in a data series p was established as $T_{p}$, where p represents the recording index. The ESN calculation was based on the data series in U, yielding output states of set X of data series $X_{p}$, where each matrix $X_{p}$ cumulated the RC output state vectors from all the training data series time step indices t_i_:(7)$$X:\{X_{p}=\left[x_{p}\left(t_{1}\right),x_{p}\left(t_{2}\right),\ldots ,x_{p}\left(t_{i}\right),\ldots ,x_{p}\left(t_{T_{p}}\right)\right]|\forall 1<p<N_{\mathrm{TS}}\}$$

Next, a set Y of N_TS_ was calculated for the goal data time series of both GED and GRF as follows:(8)$$Y:\{Y_{p}^{t}=\left[y_{p}\left(t_{1}\right),y_{p}\left(t_{2}\right),\ldots ,y_{p}\left(t_{i}\right),\ldots ,y_{p}\left(t_{T_{p}}\right)\right]|\forall 1<p<N_{\mathrm{TS}}\}$$
where $Y_{p}^{t}$ is the matrix cumulating the goal data vectors on all the signal indices time step. The RR method was used to establish the error minimum between states X and goal values $Y^{t}$, as Equation (9):(9)$$W_{\mathrm{out}}=Y^{t}X^{t}{\left({\mathrm{XX}}^{T}+\gamma I\right)}^{-1}$$
where γ is the regularization parameter to avoid overfitting, and I is an identity matrix of size N × N. The ${\mathrm{XX}}^{T}$ in Equation (10) was determined based on the sum of the matrices of the pre-multiplied output states. For each goal data, ${\mathrm{XX}}^{T}$ individually was introduced.
(10)$${\mathrm{XX}}^{T}=\sum_{p=1}^{N_{\mathrm{TS}}}X_{p}X_{p}^{T}$$

Next, the $Y^{t}X^{T}$ matrix was defined based on the sum of the pre-multiplied matrices of Xp and Yp on the data series of goal data as shown in Equation (11).
(11)$$Y^{t}X^{T}=\sum_{p=1}^{N_{\mathrm{TS}}}Y_{p}^{t}{\;X}_{p}^{T}$$

Hence, an output matrix, W_out_, was formed and applied in consistent operation mode on all training data. Subsequently, in operation mode, the predictions were made by uniformly applying the ESN function to the new data (Equation (12)).
(12)$$y\left(t\right)={\;W}_{\mathrm{out}}\mathrm{ESN}\left(u\left(t\right)\right)=W_{\mathrm{out}}x\left(t\right)$$

Figure 6 illustrates this method in training and operation modes.

#### 2.5.2. Kernel Training

With the kernel method, the GED task allowed for the evaluation of the temporal parameters. Next, the several W_out_ matrices were limited to a single range of values for these temporal parameters. Reducing the range allowed an RC training to be more specific for GRF prediction, which reduced generalizations. Hence, the RC was focused on each individual’s condition according to the metrics established as weighting parameters.

Weighting variables

The temporal parameters were used as weighting variables. W_out_ matrices were specifically trained with a greater weighting on a reduced range of these temporal parameters. The obtained parameters were integrated into an RC parallel task. This weighting method used the RC temporal parameters obtained from the GED task. Three temporal parameters were tested using this method, based on the HS and the TO gait events prediction: (1) the stance duration (duration between consecutive events of the HS and the FF), (2) the gait cycle duration (duration between two HS), and (3) the ratio between the stance phase and the gait cycle.

2.Ridge regression training with weighting

The outputs weight matrices ($W_{{\mathrm{out}}_{1}},W_{{\mathrm{out}}_{2}},\ldots ,W_{{\mathrm{out}}_{k}}$) were created using the principle of weighting the output weights [37]. Each matrix was “specialized” to predict a single portion of the data among the whole. This principle is an extension of the RR method allowing for the assigning of a greater importance to a time step portion in a sequence [37]. In practice, this weighting was applied by adding a weighting matrix $P_{\mathrm{pk}}$ in the calculation of RR on the X and Y training data sets, as follows:(13)$$W_{{\mathrm{out}}_{k}}=RR(X,Y,P_{pk})=[\sum_{p=1}^{N}Y_{p}^{t}P_{pk}X_{P}^{T}]([\sum_{p=1}^{N}X_{p}P_{pk}X_{P}^{T}]+\gamma I)$$
where the index k represents the parameters index to use for a given $W_{{\mathrm{out}}_{k}}$ matrix. The strictly diagonal matrix $P_{\mathrm{pk}}$, of size $T_{p}\times T_{p}$ corresponds to a matrix of weights as shown in Equation (14).
(14)$$P_{\mathrm{pk}}=\left[\begin{array}{ccccc} p_{\mathrm{pk}}\left(t_{1}\right) & 0 & 0 & 0 & 0\\ 0 & p_{\mathrm{pk}}\left(t_{2}\right) & 0 & 0 & 0\\ 0 & 0 & \ddots & 0 & 0\\ 0 & 0 & 0 & p_{\mathrm{pk}}\left(t_{T_{p}-1}\right) & 0\\ 0 & 0 & 0 & 0 & p_{\mathrm{pk}}\left(t_{T_{p}}\right) \end{array}\right]$$

$P_{\mathrm{pk}}$ assigns an importance value to each data sample in the matrix $W_{{\mathrm{out}}_{k}}$ estimation, with a higher significance for a higher value. For each index k, the temporal parameter value (the centre) was selected within the range of the experimental temporal parameter previously observed in the GED training. The mean value of each three temporal parameters was computed for each record of the testing set. One kernel training process was performed on each temporal parameter. The choice of the $P_{\mathrm{pk}}$ value varies for each record and for each temporal parameter.

The records having temporal parameter close to the temporal parameter centre are of high value (and become more important in the training). Records having temporal parameters that are distant from the temporal parameter centre have a low value. The distribution of the value $P_{\mathrm{pk}}$ importance followed a Gaussian function.

### 2.6. Statistical Analysis

The mean absolute error (MAE) was defined as follows:(15)$$\mathrm{MAE}=\frac{1}{{\;N}_{\mathrm{TS}}}\sum_{p=1}^{N_{\mathrm{TS}}}\left(\frac{1}{T_{p}}\sum_{i=1}^{T_{p}}\left|y_{p}\left(t_{i}\right)-y_{p}^{t}\left(t_{i}\right)\right|\right)$$
where $y_{p}\left(t_{i}\right)$ is the prediction and $y_{p}^{t}\left(t_{i}\right)$, the target value for GED and one of the GRF axes. With the kernel method, the MAE was obtained for the three zones of the full gait cycle and for the GRF active and passive peaks. One zone was determined as the total gait cycle. Two other zones counted for (0–18%) and (44–52%) of the gait cycle due to the importance of having, a maximum weight acceptance and a push off for GRF estimation, respectively [48]. For all individuals (patients and able-bodied), the active and passive peaks were happening in these two zones.

## 3. Results

### 3.1. Gait Event Detection Prediction

A box plot illustrates the results of the MAE (Figure 7) of all records for each event for the twenty-seven healthy adults (age = 36.66 ± 4.24 years old, height = 1.72 ± 0.08 m, weight = 73.11 ± 16.45 kg, SS = 1.28 ± 0.13 m/s) and for the 18 MKOA individuals (age = 65.63 ± 7.32 years old, height = 1.65 ± 0.12 m, weight = 80.51 ± 15.27 kg, SS = 0.75 ± 0.23 m/s). In able-bodied individuals, the minimum MAE of all events was found by MFT. (ML.V) in both able-bodied (mean = 27.19, SD = 5.65 ms) and MKOA (mean = 39.58, SD = 6.33 ms) individuals. For identifying only HS and TO events, the IMU on the TS yielded the best MAE for able-bodied (mean = 15.88, SD = 1.50) and MKOA individuals (mean = 33.96, SD = 5.78) (Table 4 and Table 5, respectively). In able-bodied individuals, for TS, H, MM, and MK, the best axis combinations were AP-V, ML-V, AP-V and AP-ML-V, respectively (Figure 7). For MKOA individuals, the best axis combinations were AP-ML-V, ML-V, AP-ML-V and AP-ML-V, respectively (Figure 7).

### 3.2. Ground Reaction Force Prediction

Figure 8, Figure 9 and Figure 10 show the MAE for the prediction of AP, ML and V GRF components, respectively. For each method and interval, the location/axis combination having the lowest MAE was found. The best predictor signal combinations appear in Table 6, Table 7 and Table 8 for the AP, ML and V directions, respectively. These tables show the value of the MAE for these combinations, and the distance of this minimum value of the MAE compared to the other possible location/axis combinations. This distance is represented as standard deviations between the minimal MAE found and the average of the MAEs of all the possible location/axis combinations.

When comparing the results of the standard and the kernel methods, the standard method for GRF prediction (AP, ML, V) for all locations on the leg always yielded a greater minimum MAE than the kernel training method (Figure 8, Figure 9 and Figure 10). Moreover, the errors obtained with the kernel training method are very similar for each weight variable. Hence, the segmentation data for these three parameters is alike, while combining some locations and axes of acceleration yielded slightly better than average errors (e.g., TS. (AP.V) in the stance-proportion approach compared to TS.V in cycle duration for (10, 18) zones).

For multidimensional GRF prediction, sensor location on the TS showed the lowest MAE for both healthy (72.2%) and MKOA (41.7%) individuals (Figure 11). The combinations of location or axes that yielded the best results for healthy and MKOA individuals were (TS/AP, TS/V and MM/V) and (H_AP, TS_V and MM_AP), respectively (Figure 11). However, for both groups, the TS/AP and TS/V often appeared as the best combinations, with lower MAE values (Table 6, Table 7 and Table 8). Even without the best combination, its MAE always remained low and thus very close to the best combination (Figure 8, Figure 9 and Figure 10).

Accordingly, as Table 6 and Table 7 illustrates, the best prediction for healthy individuals for the 44–52% interval was obtained with the MM/AP.ML location/axes. However, Figure 8 shows that the TS/AP prediction was nearly the second best for this interval, with an almost identical error to that of the MM/AP.ML. As for the first interval prediction (10–18%, 2nd column) in vertical direction, the best prediction was for the TS/V (Table 8), with a close accuracy prediction compared to TS/AP.V and TS/AP, respectively. For GRF.AP predictions for healthy individuals, the best combination was TS/AP, whereas for the GRF.V and ML prediction, it was TS/V. Also, when comparing the results in Figure 7 and Figure 8, the TS/AP results were always close to the best. Indeed, the TS/AP combination location generally yielded the lowest MAE for predicting multidimensional GRF for healthy individuals. The same analysis for the MKOA individuals (Figure 8, Figure 9 and Figure 10) showed that the combination TS/AP and TS/V always yielded the lowest or almost lowest MAE.

## 4. Discussion

This paper presents sensor locations for GED and multidimensional GRF predictions using ESN for able-bodied and MKOA individuals. Thus, three aspects were investigated: (1) determining the best location for predicting GED and GRF between the top of the shoe, the heel, above the medial malleolus, the middle and the front of the tibia and the medial of shank close to the knee joint; (2) comparing the ESN system with previous algorithms; and (3) comparing the ESN (standard vs. kernel training approach) performance for predicting these biomechanical metrics.

### 4.1. Determining the Best Location

For sensor location, the TS-located IMU performed best at predicting multidimensional GRF and GED using ESN for both able-bodied and MKOA individuals. For GRF.AP prediction using the kernel method (stance proportion approach in an interval of [0, 100]), the TS.AP presented the lowest MAE for able-bodied (3.2% of BW) and MKOA (2.8% of BW) individuals. The same GRF.ML prediction analysis showed that the TS.AP yields the lowest MAE for able-bodied (1.6% of BW) and MKOA (1.4% of BW) individuals. As for GRF.V prediction, TS. (AP. V) and H.V yielded the lowest MAE for able-bodied (12.6% of BW) and MKOA (11.2% of BW) individuals, respectively. The same approach showed that TS had the lowest MAE for multidimensional GRF curve prediction for able-bodied (72.2%) and MKOA (41.7%) populations (Figure 11), or in the case of not having the minimum, with only a negligible difference with it (Figure 8, Figure 9 and Figure 10). In addition, the multidimensional GRF prediction results (interval of [0, 100]) were good compared to the results of Karatsidis et al. [19] and Ohtaki et al. [32]. Karatsidis et al. [19] used a set of 17 IMUs (3D Acc, 3D Gyro and 3D Mag) to estimate multidimensional GRF in gait for 11 healthy individuals. This method, even successful (lateral component RMSE = 13.1%, frontal component RMSE = 29.6% and transverse component RMSE = 18.2%), still required full-body kinematics derived from motion equations that need information obtained from inertial motion capture (IMC) system for estimating multidimensional GRF. Ohtaki et al. [32] proposed a simpler model that used three one-dimensional IMUs placed on the distal position of the shank, the thigh and the lumbar with hook and loop fastener straps for six healthy individuals. An inverse dynamic approach allowed for the obtaining of the vertical and horizontal GRF components. The maximum RMSE of 31% of the BW for vertical and 7.6% of the BW for the horizontal GRF components was achieved. Differences in GRF prediction accuracy (0, 100) could be due to the different sensor locations and methodologies.

Besides the GRF curve, the GRF peaks prediction, including the passive (weight acceptance) and active (push off), are both critical for clinical applications. The passive peak results from the impact of the foot with the ground, while the active is caused when the foot pushes off the ground. The magnitude and timing of these peaks represent the loads experienced by muscles and joints and may result in OA incidence and progression [6,48]. Thus, the sensor location for GRF prediction in the (10, 18) and (44, 52)% intervals of the gait cycle were the passive and active GRF peaks that happened for all able-bodied and MKOA individuals is crucial. The results of this study suggest that TS.AP is the best location based on the lowest MAE in these two intervals (Table 6, Table 7 and Table 8). Indeed, the lowest MAE was lower than 4.3% and 15.3% for horizontal and vertical GRF components for both able-bodied and MKOA individuals. These results were good compared to Neugebauer et al. [6] and Thiel et al. [33]. Neugebauer et al. [6] estimated the peak GRF.V in walking and running using 3D Acc of an IMU placed on the right hip iliac crest. They recruited thirty-nine healthy adults and used statistical modelling for predicting the biomechanical metric. The results yielded an average absolute percentage difference of 8.3% between actual and predicted GRF.V peak. They also concluded that the acceleration of the hip does not correctly represent the GRF. Thiel et al. [33] used IMUs (3D Acc, 3D Gyro and 3D Mag) on each shank for estimating the GRF peak in sprint running. The average percentage error between the GRF peak estimated via linear modelling and its true value varied between 3.29% and 33.32% in the three elite sprinters recruited. However, these studies neither investigated GRF prediction regarding the best sensor location nor considered GED.

For GED based on one sensor, the ESN developed in this study appeared as a reliable tool to successfully predict up to five gait events (HS, HP, FF, TP and TO) in able-bodied and MKOA individuals. For these events’ prediction, MFT. (ML.V) appeared as the most accurate location for GED in able-bodied and MKOA individuals (MAE lower than 50 ms for all events). However, for identifying HS and TO events only, TS.AP yielded the best results in this study. For able-bodied individuals, TS.AP for HS and TO event detection had an MAE of 15 and 17 ms, respectively. However, for MKOA, these values were 30 and 38 ms, respectively. TS. AP also predicted all gait events (HS, HP, FF, TP, TO) successfully for both populations. The MAE found via TS.AP for all these events, except for FF, were lower than 56 and 66 in able-bodied and MKOA individuals, respectively. The prediction performance varied depending on the event classes. Identifying FF was a harder task when using the Acc. Acc characteristics can explain the prediction performance variability. HS and HP events occur during the impact when the foot meets the ground. It causes the largest acceleration peak in the walking cycle. Even manually, precisely estimating these two events by finding the highest peak is easy because it is obvious in the gait cycle. FF and HO events occur when the foot rests on the ground without movement. Therefore, acceleration has no peak in this part of the walking cycle. Identifying these events was harder. Events prediction in the loading response phase can therefore be considered a more complex challenge for an ESN. The TO event happens at the end of the gait cycle where this event creates an acceleration peak, similar to HS. Therefore, these events also vary depending on the shape and the amplitude of the signal caused by the walking condition or in case of MKOA. Nevertheless, the findings of this study for GED prediction were improved compared to those in the studies of Ohtaki et al. [32] and Romijnders et al. [49].

Ohtaki et al. [32] detected gait event based on the frequency analysis of the radial acceleration of the IMU placed on the middle of the tibia. The average difference of the detected events for H.S, T.P and T.O in a cadence of 60 steps/min was 0.12, 0.04 and −0.19 (s). The study conducted by Romijnders et al. [49] used the angular velocity of the two IMUs installed on both shanks to determine HS and TO events in healthy older adults and pathological individuals. In healthy older adults, they found the MAE of HS and TO to be 32 and 31 ms in straight-line walking. In pathological individuals (Parkinson’s disease), they found 33 and 40 ms for HS and TO, respectively. However, using the angular velocity of IMU for GED prediction is not recommended since there is a risk of field magnetic disturbances. The difference in their findings [32,50] compared to this study could be explained by both the sensor location and the methods used.

Nevertheless, a current study showed that the signal input retrieved from the sensor placed on TS with the uniaxial AP allows the RC to predict GED and GRF as accurately as possible.

### 4.2. Echo State Network Compared to Previous Algorithms

The ESN presented here was an efficient and robust algorithm for predicting GRF and GED using only the uniaxial Acc in the AP of the IMU placed on the TS. The real-time attribute and low-computational cost aspects of the algorithm developed in this study are highly valuable when implementing compact IMUs that generally have low computational power. This study also considered different gait velocities to expand ESN applications by preserving the ecological environment characteristics. Indeed, expended applications allow for the maintaining of the ESN accuracy for GED and GRF prediction for both able-bodied and MKOA individuals in future applications outside the lab. With the training data variability (e.g., different gait velocities), the MAE increased slightly compared to the previous test on a smaller population, showing that the ESN generalization capacity requires a small precision trade-off [42].

For training, different ESN approaches (kernel and standard methods) could be valuable depending on the application and the training data available. With specific training data using the kernel method, the ESN precision was maximized for similar testing data compared to the standard training approach. Using kernel training with temporal parameters mainly improved the prediction for cases where walking cadence was variable. The rate varied significantly with the amplitude of the GRF signals. The standard training approach prevented the anticipating or monitoring of such variations. However, the kernel method allowed a better force prediction precision when values were outside the normal range, either for very slow or very fast walks. This method may be close to some models in the literature [50], and may also make the calibration of the GRF prediction easier according to the speed used. Therefore, the kernel method would be a fine-tuning of the ESN to increase the precision for a specific application. This training method can use information related to gait, such as temporal parameters, to improve prediction results. Other gait parameters could be used in the kernel training method to improve the prediction performance. Regardless, even if the training method slightly improves the prediction, the accuracy is almost dependent on the nature of the Acc input signal.

The accelerations obtained from different sensor locations showed differences in amplitude and signal shapes, which can explain the different biomechanical metric prediction performances. However, the RC method with TS.AP allows a good prediction of multidimensional GRF and GED. The accuracy with TS.AP of GRF prediction is of the order of 12.6% of the BW for vertical forces, 3.2% of the BW for AP forces, and 1.6% of the BW for ML forces in the (0, 100) interval of the gait cycle. RC also showed robustness in GRF prediction with a minimum error distance varying between 1.8 and 2.6 standard deviations from the mean. For (10, 18) and (44, 52) % of the gait cycle, when passive and active peaks happen, the MAE was lower than 4.3% and 15.3% for horizontal and vertical GRF components. Compared to Ngoh et al. [24], this prediction is an intermediate method to estimate the GRF.V using TS.AP. Indeed, they suggested a two-layer feed-forward NN (FNN) to estimate the GRF vertical component (GRF.V) with AP IMU Acc on the TS for running in seven healthy young males. The average resulting RMSE is less than 1.7% of the BW for GRF prediction. The average error obtained for impact and the GRF active force ranged between 10% and 18% of the BW. However, it is easier to predict the GRF for running compared to walking. The double support in walking restricts the GRF prediction, hence, studies like Karatsidis et al.’s [19] even used an extra function (smooth transition function) in the double support phase of the walk to improve the prediction accuracy. However, they still achieved poor accuracy (the lateral component RMSE = 13.1%, frontal component RMSE = 29.6%, and transverse component RMSE = 18.2%). In addition, the database for FNN used in the study of Ngoh et al. [24] was limited to seven individuals, while the ESN database of this study was larger with more participants (27 able-bodied and 18 MKOA individuals) and more gait cycles (about 350 gait cycles), and thus such variability in the training data was not considered over previous literature [23,24].

### 4.3. Limitations and Future Work

The current work remains limited. The findings are meant for an integrated GED and GRF task and sensor location and are thus limited to the shank and the foot segments only. While some locations such as L5 appeared as very promising for GRF prediction only, this option could be tested in future works. In addition, the results are only based on the data collected in a lab setting condition and it is not always enough to only consider the characteristics of the ecological environment (e.g., variability in gait speed [9]). Furthermore, the GRF prediction performance with the RC method is still insufficient for clinical analysis applications. Indeed, the force variations are generally of the order of 100-150 N between able-bodied and MKOA individuals [50] in the stance phase of the gait cycle. This sums up to about 15 to 20% of the BW of the total amplitude of the vertical ground forces in the cycle steps. The RC can yield an error of up to 25% in high amplitude zones of vertical forces on the ground. The error of the model is therefore too important to show the subtle variations that a correction might bring.

Improvements in the RC algorithm can still reduce this error. The FNN entries in the Leporace et al. [23] study could be implemented in the RC to improve its performance. The Leporace et al. [23] FNN included acceleration values of derivatives and integrals from the beginning of the gait cycle. In their method, the integral value is reset at each cycle based on the detection of an HS event. For RC, it will then be possible to improve the prediction performance by adding a counter of an integrated value of the input acceleration as another input. However, since the goal of this study is to eventually migrate to the hardware reservoir computing Micro Electronic Mechanical Systems (MEMS) technology, this modification is hard to make in the RC- MEMS version of tank computing. A second modification would be to define a matrix of output weights multiplied at the RC output states for several time steps for a single prediction y(t) of the RC. This improvement of the RC, unlike the first, is compatible with the MEMS-RC, since the digital modifications would be applied only to the output states of the RC. Numerically, this compares to applying a filter to the output data adjusted from the RR regression. Finally, having data acquired over multiple walking patterns including the different gait retraining patterns (e.g., foot progression angle, the trunk lean or the knee medialization), that are typically retrieved from a larger population sample of MKOA individuals would be highly useful to train the ESN model used in this study and to validate its precision to detect GRF even with possible variations induced by the correction of the gait pattern for a future clinical application.

## 5. Conclusions

This paper investigates the best sensor location for GED and GRF prediction scenarios in able-bodied and MKOA individuals using AI in gait. An ESN approach (with kernel and standard training methods) used combinations of 3D Acc input in both scenarios to find the best location or input Acc in both populations. The results showed an IMU located on the TS via AP Acc input best estimates these biomechanical metrics accurately. The results showed that the RC using only one sensor could produce an acceptable prediction for both tasks, unlike other methods developed in the literature that are effective for one prediction task only [19,26]. The findings of this study open the way for gait retraining applications in the ecological environment using only one IMU for providing feedback based on estimated biomechanical metrics (e.g., GRF prediction, stride time estimation, stance and swing phase duration without relying on a gait laboratory). Also, the ESN used in this study aims to lower the sensor computational cost and energy consumption to help its implementation in wearables. An on-board wearable algorithm will decrease the device consumption by reducing the data communications that the system must perform [51]. ESN can lower the energy consumption of the sensor by reducing the memory and the processing time needed to run, which is another indirect indicator of the algorithm’s needs with regard to energy [52,53]. However, previous research overlooked this matter [54,55,56] despite its importance. 

Nevertheless, wearables to monitor and guide treatment must be autonomous, and this is a promising approach for later out-lab applications. The ESN used in this study is inexpensive in calculation and in memory. Also, the best sensor location/predictor Acc signal will increase the accuracy and reliability of the prediction and reduce the system computational cost. Hence, AI-based wearable sensors will be more easily used to deliver treatments and monitor in ecological environments. This also opens the door to new AI-MEMS [36] architecture technologies for non-linear GED and GRF prediction related to biomechanical purposes that require an even lower scale factor and energy, and that can improve device use and longevity.

## Figures and Tables

**Figure 1 ijerph-20-03120-f001:**
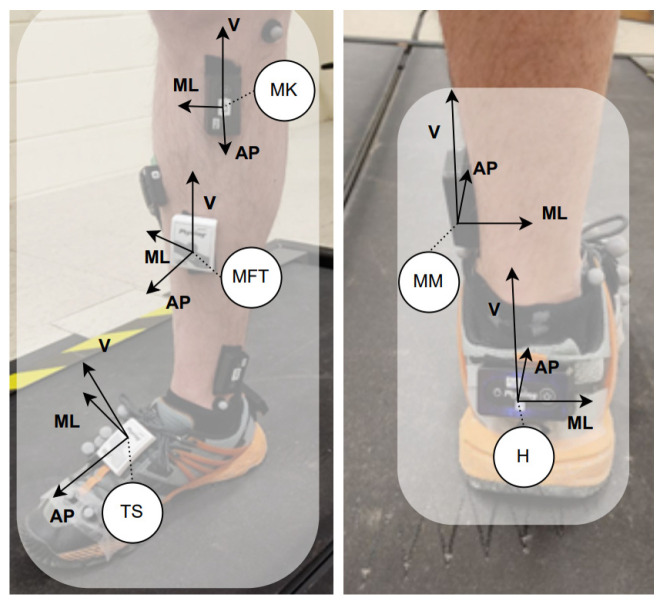
IMUs location and orientation.

**Figure 2 ijerph-20-03120-f002:**
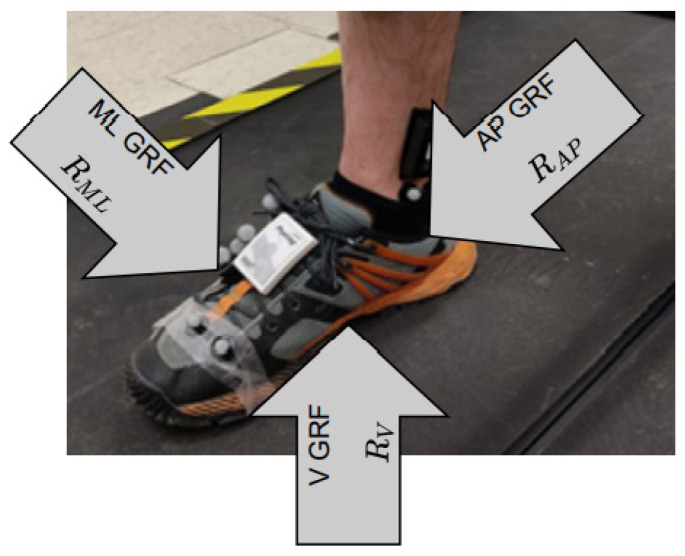
Ground reaction force orientation (Bertec, 1000 Hz).

**Figure 3 ijerph-20-03120-f003:**
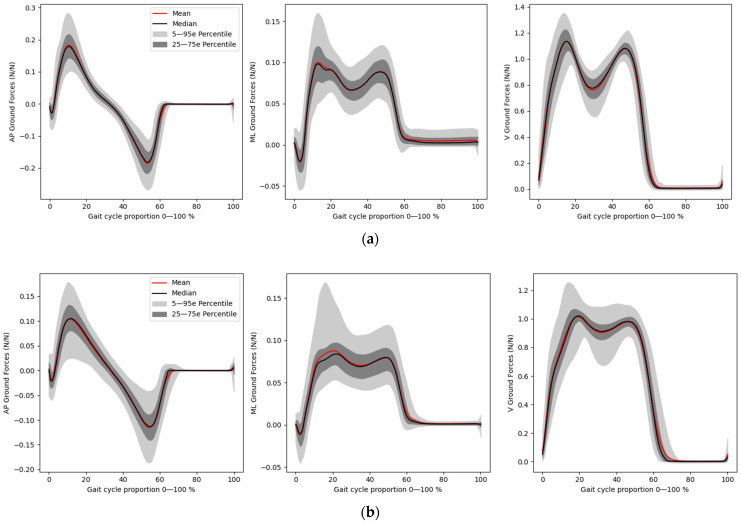
Ground reaction force of (**a**) healthy and (**b**) MKOA individuals.

**Figure 4 ijerph-20-03120-f004:**
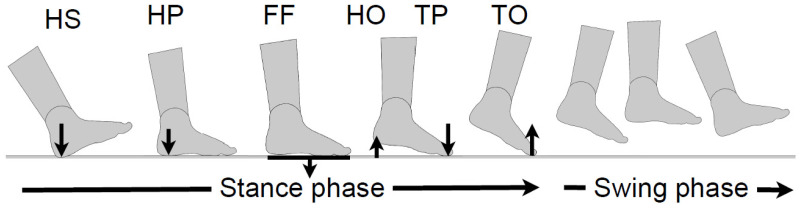
Gait events investigated in the stance phase.

**Figure 5 ijerph-20-03120-f005:**
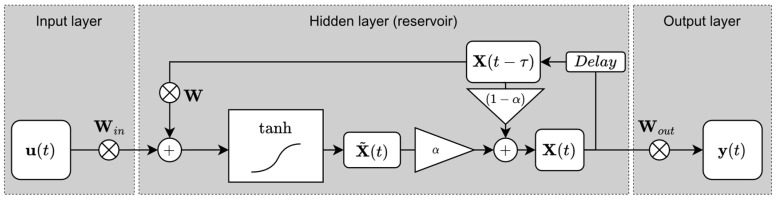
Architecture of the ESN (the cross symbol represents matrix multiplication).

**Figure 6 ijerph-20-03120-f006:**
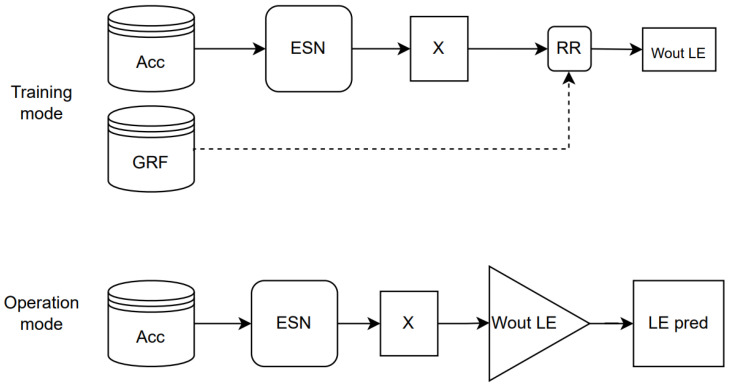
ESN method.

**Figure 7 ijerph-20-03120-f007:**
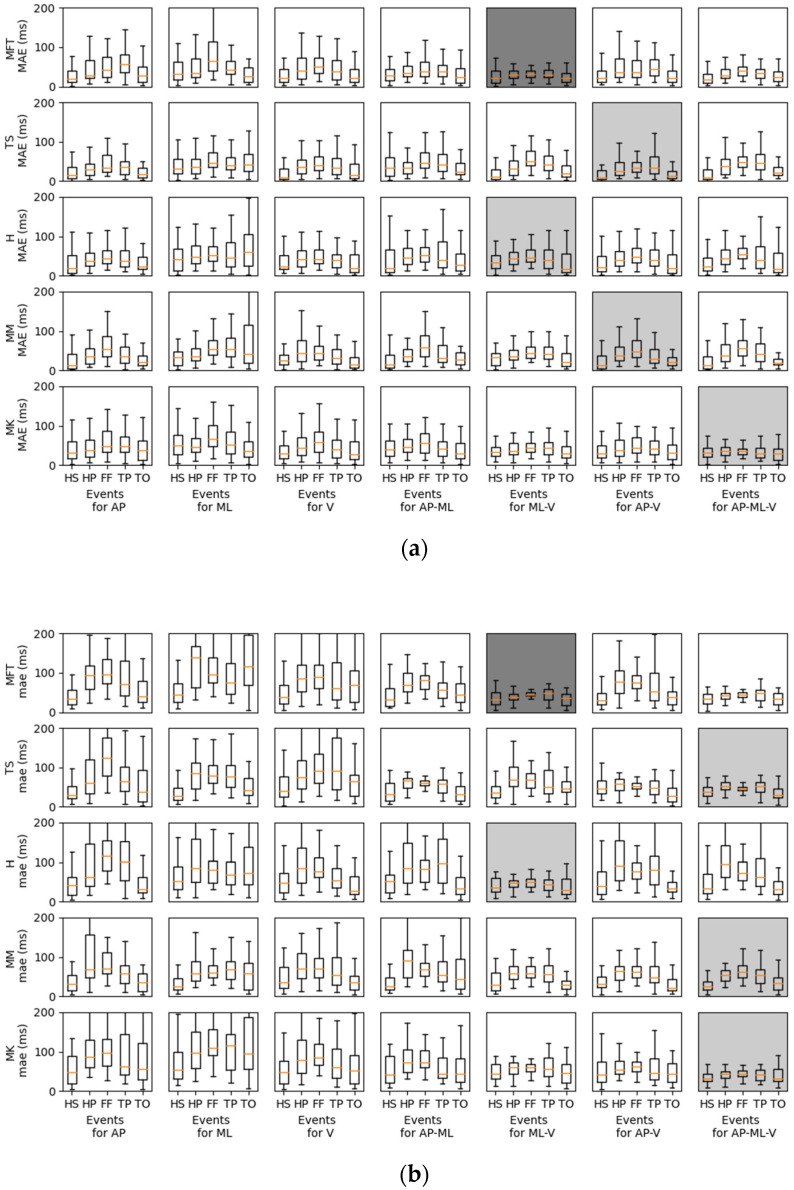
GED prediction for the (**a**) healthy and (**b**) MKOA individuals. For each row, the graph represents the MAE for one sensor location (i.e., MFT, TS, H, MM, MK). The results having the smaller MAE over all events and records for each location appear in light grey. The results with the smallest MAE over all the combinations and locations appear in dark grey.

**Figure 8 ijerph-20-03120-f008:**
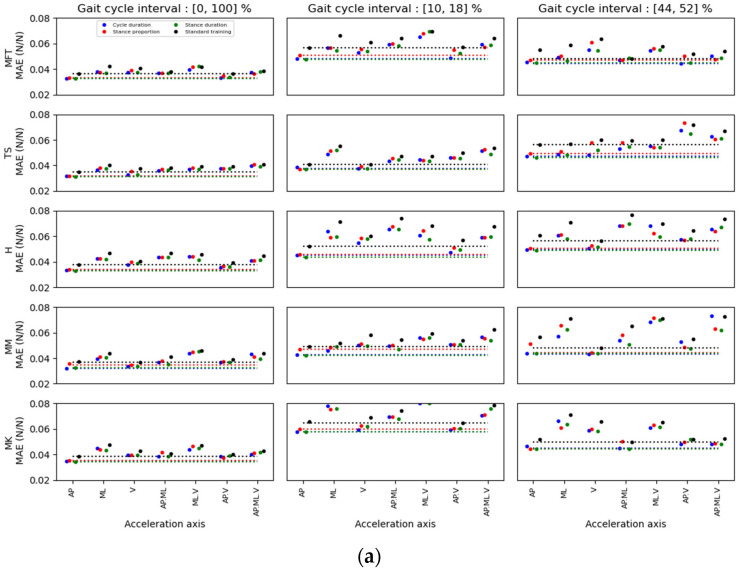
MAE results for the AP forces estimation for different sensor location and axis combinations in (**a**) able-bodied and (**b**) MKOA individuals. For each acceleration axis, four MAE points were identified: the first three were obtained with the kernel training method. The coloured dots represent the error after the kernel training with different weighting variables. Blue dots represent the cycle duration, red dots represent the duration proportion of the stance phase, and green dots represent the stance phase duration. Black dots represent the MAE for a standard training method. The dotted lines represent the minimum values found for each weighting variable and the standard method.

**Figure 9 ijerph-20-03120-f009:**
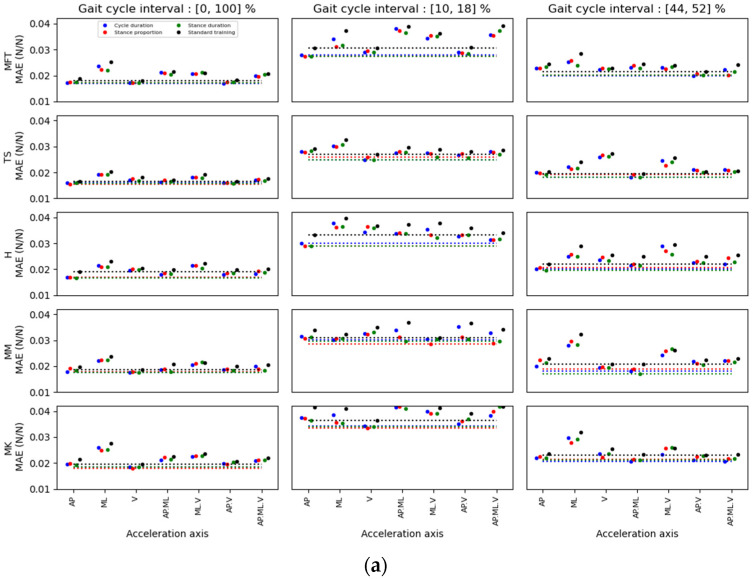
MAE results for the ML forces estimation for sensor locations and axis combinations in (**a**) able-bodied and (**b**) MKOA individuals. For each acceleration axis, four MAE points were identified: the first three were obtained with the kernel training method. The coloured dots represent the error after the kernel training with different weighting variables. Blue dots represent the cycle duration, red dots represent the duration proportion of the stance phase, and green dots represent the stance phase duration. Black dots represent the MAE for a standard training method. The dotted lines represent the minimum values found for each weighting variable and the standard method.

**Figure 10 ijerph-20-03120-f010:**
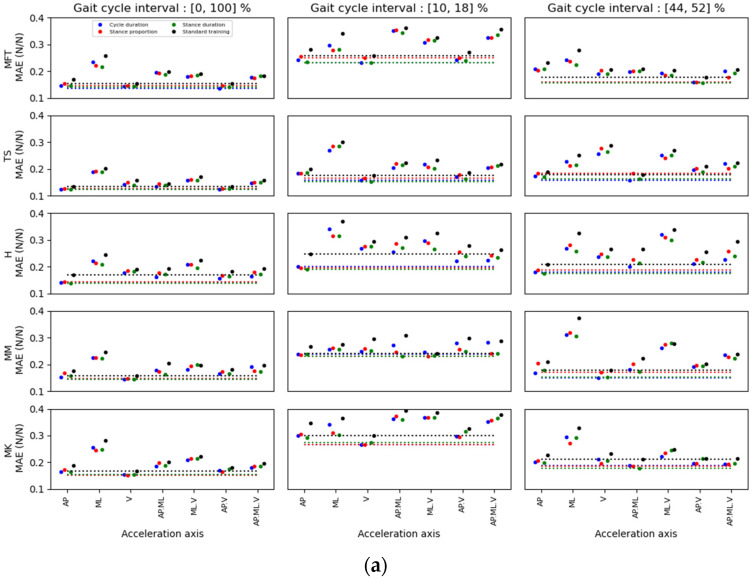
MAE results for the V forces estimation for different sensor locations and axis combinations in (**a**) able-bodied and (**b**) MKOA individuals. For each acceleration axis, four MAE points were identified: the first three were obtained with the kernel training method. The coloured dots represent the error after the kernel training with different weighting variables. Blue dots represent the cycle duration, red dots represent the duration proportion of the stance phase, and green dots represent the stance phase duration. Black dots represent the MAE for a standard training method. The dotted lines represent the minimum values found for each weighting variable and the standard method.

**Figure 11 ijerph-20-03120-f011:**
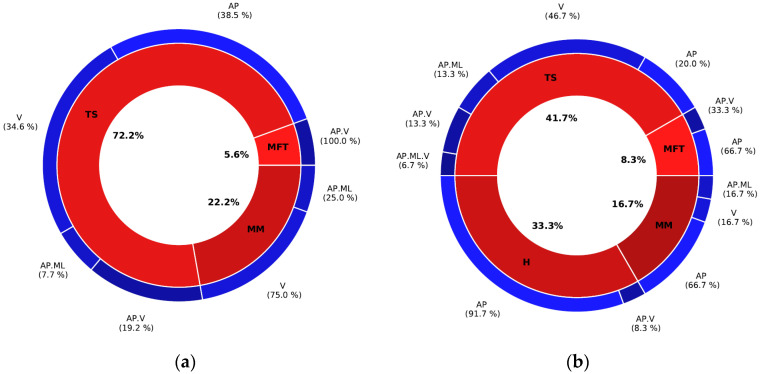
The locations and axes having the minimum prediction error for three GRF orientations in (**a**) able-bodied and (**b**) MKOA individuals, and according to the three intervals of the cycle of steps.

**Table 1 ijerph-20-03120-t001:** Participants’ demographic information.

Characteristics	Healthy (*n* = 27)	MKOA (*n* = 18)
Age (y)	36.66 ± 4.24	65.63 ± 7.32
Gait velocity (m/s)	1.28 ± 0.13	0.75 ± 0.23
Height (m)	1.72 ± 0.08	1.65 ± 0.12
Mass (kg)	73.11 ± 16.45	80.51 ± 15.27

**Table 2 ijerph-20-03120-t002:** Defining u(t) axes of the input vector.

Axes	AP	ML	V	AP-ML	ML-V	AP-V	AP-ML-V
$u\left(t_{i}\right)$	$\left[\begin{array}{c} 1\\ a_{AP}\left(t_{i}\right) \end{array}\right]$	$\left[\begin{array}{c} 1\\ a_{ML}\left(t_{i}\right) \end{array}\right]$	$\left[\begin{array}{c} 1\\ a_{V}\left(t_{i}\right) \end{array}\right]$	$\left[\begin{array}{c} 1\\ a_{AP}\left(t_{i}\right)\\ a_{ML}\left(t_{i}\right) \end{array}\right]$	$\left[\begin{array}{c} 1\\ a_{ML}\left(t_{i}\right)\\ a_{V}\left(t_{i}\right) \end{array}\right]$	$\left[\begin{array}{c} 1\\ a_{AP}\left(t_{i}\right)\\ a_{V}\left(t_{i}\right) \end{array}\right]$	$\left[\begin{array}{c} 1\\ a_{AP}\left(t_{i}\right)\\ a_{ML}\left(t_{i}\right)\\ a_{V}\left(t_{i}\right) \end{array}\right]$

**Table 3 ijerph-20-03120-t003:** Echo State Network Hyperparameters.

Hyperparameters	Symbol	Value
Number of nodes	N	100
Leaking rate	α	0.1053
W spectral radius	ρ	0.7471
W sparsity	ρ	0.21
Regularization parameter	γ	1 × 10^−6^

**Table 4 ijerph-20-03120-t004:** GED estimation for able-bodied individuals.

Axis	Event	MAE (ms) by Location				
		MFT	TS	H	MM	MK
AP	HS	19.49	14.84	18.92	12.18	30.80
	HP	28.10	28.68	37.38	35.13	37.43
	FF	42.24	34.63	44.61	54.03	47.24
	TP	55.71	36.72	37.81	34.68	48.29
	TO	27.06	16.92	23.51	22.05	36.94
ML	HS	31.92	31.88	41.08	33.06	48.97
	HP	34.77	36.82	48.63	35.42	45.68
	FF	63.75	45.14	52.76	54.17	66.13
	TP	42.92	40.81	45.37	54.69	51.53
	TO	25.63	42.57	60.14	40.78	35.89
V	HS	21.43	9.58	23.92	24.79	30.26
	HP	40.81	34.86	42.03	43.65	43.06
	FF	51.13	39.72	41.67	43.39	57.50
	TP	37.92	33.50	38.82	31.06	40.14
	TO	21.56	15.78	19.05	15.92	27.56
AP-ML	HS	27.42	34.59	18.33	14.38	39.00
	HP	34.50	34.67	46.52	36.28	46.52
	FF	38.17	46.39	51.79	58.42	55.15
	TP	37.66	42.37	39.19	32.26	40.79
	TO	23.75	22.68	27.06	27.34	28.64
ML-V	HS	20.63	10.22	34.26	33.00	34.32
	HP	30.78	32.70	44.70	36.39	35.81
	FF	32.36	49.31	46.91	42.98	43.68
	TP	30.71	40.97	39.86	41.39	42.63
	TO	21.49	19.74	17.57	21.70	29.73
AP-V	HS	22.30	9.53	21.31	13.78	29.32
	HP	36.89	24.87	40.00	37.43	38.38
	FF	36.58	33.97	47.36	47.88	42.97
	TP	44.31	33.03	38.82	30.34	42.50
	TO	21.03	13.75	18.30	21.46	31.29
AP-ML-V	HS	17.74	9.66	23.24	12.66	31.00
	HP	27.76	36.96	42.94	37.64	36.08
	FF	40.34	48.24	53.75	56.56	35.00
	TP	33.71	45.56	39.75	42.70	28.82
	TO	24.27	21.91	16.90	18.33	28.89

**Table 5 ijerph-20-03120-t005:** GED estimation for MKOA individuals.

Axis	Event	MAE by Location				
		MFT	TS	H	MM	MK
AP	HS	33.10	29.88	42.39	31.79	47.30
	HP	93.97	59.85	62.73	69.03	85.73
	FF	95.80	123.33	116.21	70.15	96.25
	TP	70.00	65.40	100.54	58.44	62.68
	TO	41.03	38.05	32.36	35.32	55.5
ML	HS	44.19	28.39	52.02	24.46	54.29
	HP	138.68	85.12	84.52	57.68	96.96
	FF	94.63	78.18	79.83	60.43	110.18
	TP	75.00	76.53	68.79	68.39	116.00
	TO	115.65	42.39	71.70	57.96	95.65
V	HS	38.07	40.86	47.83	35.81	46.73
	HP	84.52	75.31	84.11	71.43	79.00
	FF	89.50	92.17	75.85	70.19	84.55
	TP	59.86	90.45	54.71	54.13	60.98
	TO	68.91	64.07	26.62	35.89	52.59
AP-ML	HS	30.97	31.76	51.22	24.58	41.14
	HP	68.57	66.79	84.71	91.55	71.55
	FF	81.38	60.00	83.10	68.00	72.08
	TP	55.65	57.50	96.28	54.00	43.72
	TO	43.50	32.58	34.35	43.91	42.93
ML-V	HS	32.29	36.63	35.88	28.87	43.14
	HP	38.31	69.60	47.58	59.13	59.50
	FF	44.40	68.33	49.24	58.70	59.52
	TP	47.59	50.42	44.88	55.33	55.45
	TO	35.34	46.33	30.48	29.15	45.64
AP-V	HS	30.33	46.07	39.41	31.29	42.29
	HP	77.50	57.60	91.25	65.00	54.60
	FF	75.69	52.50	76.90	62.94	61.88
	TP	53.39	48.37	80.41	48.38	46.13
	TO	38.83	27.80	33.71	22.20	44.00
AP-ML-V	HS	33.43	38.39	34.17	25.57	30.83
	HP	42.79	53.28	95.88	54.05	44.04
	FF	44.13	46.67	73.41	62.73	44.05
	TP	48.63	52.13	62.18	53.81	41.12
	TO	34.80	30.15	32.29	32.50	31.90

**Table 6 ijerph-20-03120-t006:** Summary of the best location and all axis combinations associated with GRF estimation in AP direction.

Method	Interval (%)	Mean Absolute Error		Ave p/r Dist (1/σ)		Best Location		Best Axis	
		Healthy	MKOA	Healthy	MKOA	Healthy	MKOA	Healthy	MKOA
Cycle duration	(0, 100)	0.032	0.025	−1.679	-2.347	TS	MM	AP	AP
	(10, 18)	0.037	0.029	−1.742	-2.271	TS	TS	V	V
	(44, 52)	0.043	0.035	−1.400	−1.776	MM	MM	V	AP
Stance proportion	(0, 100)	0.032	0.028	−2.077	−2.037	TS	TS	AP	AP
	(10, 18)	0.037	0.031	−1.934	−2.804	TS	TS	AP	V
	(44, 52)	0.044	0.036	−1.549	−1.750	MM	H	V	AP
Stance duration	(0, 100)	0.031	0.026	−1.829	−2.429	TS	MM	AP	AP
	(10, 18)	0.037	0.029	−1.731	−2.498	TS	TS	AP	V
	(44, 52)	0.043	0.034	−1.338	−1.898	MM	MM	V	AP.ML
Standard training	(0, 100)	0.035	0.031	−1.829	−1.860	TS	MM	AP	AP
	(10, 18)	0.041	0.039	−1.851	−2.483	TS	TS	AP	V
	(44, 52)	0.048	0.041	−1.675	−1.648	MM	MM	V	V

**Table 7 ijerph-20-03120-t007:** Summary of the best location and all axis combinations associated with GRF estimation in ML directions.

Method	Interval (%)	Mean Absolute Error		Ave p/r Dist (1/σ)		Best Location		Best Axes	
		Healthy	MKOA	Healthy	MKOA	Healthy	MKOA	Healthy	MKOA
Cycle duration	(0, 100)	0.016	0.014	−1.490	−1.577	TS	H	AP.V	AP
	(10, 18)	0.025	0.023	−1.942	−1.753	TS	TS	V	V
	(44, 52)	0.018	0.014	−1.680	−1.618	TS	TS	AP.ML	AP.ML
Stance proportion	(0, 100)	0.016	0.014	−1.813	−1.750	TS	TS	AP	AP
	(10, 18)	0.026	0.020	−1.582	−2.214	TS	MFT	V	AP
	(44, 52)	0.019	0.015	−1.510	−1.568	MM	TS	AP.ML	AP.ML
Stance duration	(0, 100)	0.016	0.014	−1.580	−1.604	TS	H	AP.V	AP
	(10, 18)	0.025	0.022	−1.742	−1.839	TS	H	V	AP
	(44, 52)	0.017	0.014	−2.037	−1.618	MM	TS	AP.ML	AP.V
Standard training	(0, 100)	0.016	0.016	−1.707	−1.572	TS	TS	AP	AP
	(10, 18)	0.027	0.025	−1.800	−2.021	TS	MFT	V	AP
	(44, 52)	0.019	0.016	−1.608	−1.524	TS	TS	AP.ML	AP.ML.V

**Table 8 ijerph-20-03120-t008:** Summary of the best location and all axis combinations associated with GRF estimation in V direction.

Method	Interval (%)	Mean Absolute Error		Ave p/r Dist (1/σ)		Best Location		Best Axes	
		Healthy	MKOA	Healthy	MKOA	Healthy	MKOA	Healthy	MKOA
Cycle duration	(0, 100)	0.123	0.107	−1.555	−1.559	TS	H	AP.V	AP
	(10, 18)	0.157	0.166	−1.996	−1.530	TS	TS	V	V
	(44, 52)	0.150	0.099	−1.586	−1.418	MM	H	V	AP
Stance proportion	(0, 100)	0.126	0.112	−1.744	−1.456	TS	H	AP.V	AP
	(10, 18)	0.166	0.149	−1.931	−1.830	TS	TS	V	AP.V
	(44, 52)	0.158	0.105	−1.558	−1.294	MFT	H	AP.V	AP
Stance duration	(0, 100)	0.125	0.105	−1.649	−1.539	TS	H	AP	AP
	(10, 18)	0.153	0.158	−1.992	−1.590	TS	H	V	AP
	(44, 52)	0.154	0.101	−1.513	−1.304	MM	H	V	AP.V
Standard training	(0, 100)	0.134	0.132	−1.660	−1.346	TS	H	AP.V	AP
	(10, 18)	0.176	0.189	−2.089	−1.714	TS	TS	V	V
	(44, 52)	0.176	0.122	−1.372	−1.151	MFT	MFT	AP.V	AP.V

## Data Availability

Not applicable.

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
