# Peer review of "Inertial Sensor Location for Ground Reaction Force and Gait Event Detection Using Reservoir Computing in Gait"

_ijerph, 2023, doi:10.3390/ijerph20043120_

Round 1

Reviewer 1 Report

This paper describes in detail the use of different Internal sensors for ground reaction force and gait event detection using reserve computing in gait. The overall idea is clear and has good practice. I hope there are some improvements to promote the publication of this paper.

1. Introduction should include a brief introduction to the methods used in this paper, including the results and innovations

2. Some figures in the paper are not clear and need to be improved. As shown in Figure 7, Figure 8 and Figure 9

3. The author is suggested to modify Figure 4 and use a more intuitive figure to show it. In addition, the ESN parameters need to be specified

Reviewer 2 Report

The paper presents a research related to inertial sensor location for ground reaction force and gait event detection using reservoir computing in gait. The paper is well structured and the quality of the presentation is properly conducted. Some minor improvements can be made to the quality of the graphical figures : Figure 3 (the quality is poor, and the text is not easy readable ), Quality of figures 7,8,9 must be improved.

Best regards!

Reviewer 3 Report

Inertial measurement units (IMUs) have shown promising outcomes for estimating gait event detection (GED) and ground reaction force (GRF).

This authors aim to figure out the best sensor location for GED and GRF prediction in gait using data from IMUs for healthy and medial knee osteoarthritis (MKOA) individuals.

They enrolled, 27 healthy and 18 MKOA individuals. Participants walked at different speeds on an instrumented treadmill. Five synchronized IMUs (Physilog®, 200 Hz) were placed on the lower limb (the top of the shoe [TS], the heel [H], above the medial malleolus [MM], the middle and the front of the tibia [MFT] and on the medial of the shank close to the knee joint [MK]).

To predict GRF and GED, the authors trained an artificial neural network (ANN) called a reservoir computing using combinations of acceleration signals (Acc) retrieved from each IMU. For GRF prediction, the best sensor location is the TS for 72.2% and 41.7% of individuals in healthy and MKOA populations, respectively, based on the minimum value of the mean absolute error (MAE). For GED, the minimum MAE value for both groups was for MFT, then TS.

They concluded that TS is the best sensor location for GED and GRF prediction

The study is interesting and attractive.

Main Strength points:

- The method is innovative and brings mathematics and algorithms with potential

- The introduction is very rich in information and details.

Main Weakness

- This math could be organized more compactly. It is distributed sometimes in the text and sometimes in figures (see figure 4). We suggest the use of flow charts where possible and to compact the equations into a single paragraph.

- In the final part of the introduction there is a lot of information. This is good but the reader risks getting lost and above all not understanding the true purpose of the study.

Others

1.      Minimize the use of acronyms in the abstract and insert a list

2.      Uniform the use of the verb tenses

3.      Avoid the use of we or our

4.      Introduce a clear and effective purpose in the introduction

5.      Some tables do not follow the mdpi standard

6.      Check the resolution of figures

7.      The equation numbers are not progressive

8.      Avoid the use of acronyms in the titles

Round 2

Reviewer 1 Report

The author revised the paper according to the suggestions. I suggested to publish it.